# Usability, acceptability, and self-reported impact of an innovative hepatitis C risk reduction intervention for men have sex with men: A mixed methods study

**Tamara Prinsenberg**[1,2]*, **Joël Illidge**[1], **Paul Zantkuijl**[3], **Maarten Bedert**[2], **Maria Prins**[1,2], **Marc van der Valk**[2,4], **Udi Davidovich**[1,5]

1 Department of Infectious Diseases Research and Prevention, Public Health Service of Amsterdam, Amsterdam, The Netherlands, 2 Department of Infectious Diseases, Amsterdam Infection and Immunity Institute, Amsterdam UMC, University of Amsterdam, Amsterdam, The Netherlands, 3 SOA Aids Nederland, Amsterdam, The Netherlands, 4 Stichting HIV Monitoring, Amsterdam, The Netherlands, 5 Department of Social Psychology, University of Amsterdam, Amsterdam, The Netherlands

* TPrinsenberg@ggd.amsterdam.nl

**Data Availability Statement:** The data underlying this study are available on Figshare (https://doi.org/10.6084/m9.figshare.16676872.v1).

## Abstract

Hepatitis C virus (HCV) elimination among men who have sex with men (MSM) is unlikely to be feasible without effective behavioural interventions. We developed a multilevel intervention to reduce HCV transmission among MSM in Amsterdam. The intervention includes a toolbox to facilitate risk reduction among MSM and support health care professionals in risk reduction counselling. To assess the use of the toolbox and its impact on behavior, we conducted a mixed-methods study. We collected data through online questionnaires (n = 49), and in-depth interviews with MSM at risk of HCV (n = 15) and health care professionals (n = 7). We found that the toolbox has been well received by MSM, increased awareness of HCV risks and has facilitated preventive behaviours and risk-reduction communication with peers. Professionals reported the toolbox to be a useful aid for discussions about HCV risk and risk reduction strategies with their clients.

## Introduction

Since 2000, hepatitis C virus (HCV) outbreaks among HIV-positive men who have sex with men (MSM) have been reported globally [1]. In the Netherlands, a decline in primary HCV infection and HCV reinfection has been observed in this population after the introduction of unrestricted access to direct-acting antivirals (DAAs) in 2015 [2]. Between 2008 and 2015 the HCV incidence fluctuated between 8.7 and 13.0 per 1000 person-years and significantly declined to 6.1 per 1000 person years in 2016. Between 2017 and 2019, HCV incidence fluctuated between 4.1 and 4.9 per 1000 person-years. HCV reinfection incidence also declined from 41.4 per 1000 person-years in 2016 to 11.4 per 1000 person-years in 2019, but it remained high, illustrating the ongoing HCV transmission in HIV-positive MSM in the Netherlands [2].

**Funding:** This study was performed within the MC Free consortium. MC Free is funded by grants from Gilead Sciences, AbbVie, Janssen-Cilag, Merck Sharpe & Dohme, and Roche Diagnostics. The funders had no involvement in the study design, writing of the manuscript, and decision to submit the article for publication. TP and MP report speaker fees and grants from Gilead Sciences, Merck Sharpe & Dohme, and AbbVie paid to their institute. MvdV's institute received unrestricted research grants and consultancy fees from AbbVie, Gilead, Johnson & Johnson, Merck Sharpe & Dohme, and ViiV Health Care.

**Competing interests:** The authors have declared that no competing interests exit.

HCV incidence among HIV-negative MSM is more challenging to predict as HIV-negative MSM are normally not in routine clinical care. Two systematic reviews showed a 16-to-19-fold lower HCV incidence in HIV-negative MSM compared to HIV-positive MSM including studies from 2000 to 2016 (0,4/1000 person years in HIV- negative and 6.4–7.8/1000 person years in HIV-positive MSM) [3, 4]. However, the introduction and increased uptake of pre-exposure prophylaxis (PrEP) against HIV infection, has resulted in changed patterns of sexual networks and behaviour among HIV-negative MSM [5, 6]. A recent meta-analysis, including studies from 2000 to 2019, estimated a 123-fold higher HCV incidence in HIV-negative MSM using PrEP compared to HIV-negative MSM not using PrEP and a pooled HCV incidence of 14.8 per 1000 person years in HIV-negative MSM using PrEP [7]. An additional concern is the high HCV reinfection incidence among HIV-negative MSM using PrEP. During follow-up in a PrEP demonstration project in the Netherlands an HCV re-infection incidence of 278 per 1000 person years was found [6].

HCV transmission among MSM occurs mainly through sexual contact [8]. Several studies have identified certain sexual techniques and settings as risk factors for HCV infection, including condomless anal intercourse, (unprotected) fisting, sharing of sex toys, group sex and chemsex (the use of recreational drugs immediately before and/or during sex to facilitate or enhance sexual pleasure) [8, 9]. In addition, having an ulcerative sexually transmitted infection (STI), injecting drug use, sharing of straws or other equipment for snorting drugs, sharing anal douching equipment and rectal bleeding are also associated with an increased risk of incident HCV infection [8, 9]. A study in the Netherlands found that receptive condomless anal intercourse, sharing of sex toys, group sex, anal rinsing before sex, having 10 or more sex partners in the last 6 months are strongly associated with HCV reinfection [10]. To prevent HCV (re)infections among MSM, scaling up HCV treatment and the implementation of risk reduction interventions for MSM are recommended [11]. Evidence on how to achieve HCV behavioural risk reduction in MSM is limited. Up to now, the focus of sexual risk reduction interventions has been on the use of condoms to prevent the transmission of HIV and other STIs. Yet, the prevention of HCV requires more than promoting condom use alone. Informing men at risk about the possible HCV risk factors other than anal intercourse, as well as motivating them to integrate such risk reduction strategies into their sex lives are essential. The first HCV specific sexual risk reduction intervention was developed in 2016 as part of the Swiss HCVree trial [12]: an intervention for HIV/HCV co-infected MSM provided in combination with DAA treatment and consisting of individual counselling sessions aimed at reducing sexual risk taking [13]. In 2017, we initiated the NoMoreC project, a multilevel intervention aimed at reducing HCV transmission among all MSM at risk of HCV in Amsterdam [14]. Our approach was to target MSM based on their sexual practices, rather than on their HIV or HCV-status. NoMoreC includes web-based and face-to-face components as well as an anonymous testing service. The NoMoreC website (www.nomorec.nl) provides information about hepatitis C, HCV transmission routes, risk reduction strategies, testing and treatment options, and partner notification. The face-to-face component comprises a risk reduction toolbox, training for health professionals, and providing tailored advice to sex on premises venues. NoMoreC is promoted by a voluntary community campaign team.

In this study we focus on one component of the project, the NoMoreC toolbox. The approach used to develop the toolbox was innovative, and such approach was not applied and evaluated elsewhere before. The toolbox development was based on the information motivation and behavioural skills (IMB) model [15], as IMB-based interventions have shown to be effective in preventing risky sexual behaviours among risk groups [16–18]. The NoMoreC toolbox contains practical tools, to encourage HCV risk reduction in different settings (e.g. during one-on-one sex, group sex, sex parties, sexualized drug use). Examples of these tools

are condoms for safer sex, gloves and hand disinfectant for safer fisting, single use drug equipment for safer chemsex and various cleaning and disinfection products to create safer conditions for group sex.

The Toolbox is the first comprehensive HCV risk reduction tool to provide information on HCV risks in combination with a broad range of products for risk reduction and, furthermore, it has been co-created with the gay community. We conducted a mixed-methods study among MSM at risk of HCV and health care professionals to assess the use of the toolbox and its impact on behaviour. We aimed to identify reasons for obtaining the toolbox, motives and barriers to its use, measured aspects of usability and acceptability, and assessed the toolbox's impact on HCV awareness and personal behaviour change, as self-reported by toolbox users.

## Methods

### Toolbox development and distribution

The development of the toolbox was guided by a co-creation process that included various sessions with a group of MSM. This group was formed specifically for the NoMoreC project and was composed of members of the Amsterdam gay community. It included men at risk of HCV, men who had been HCV-infected in the past, and men who had concerns about becoming infected. The IMB model [15] applied by us to the HCV context, suggests that individuals will initiate and maintain HCV preventive behaviours if they are well-informed regarding HCV, HCV transmission routes, HCV infection risks, and possible preventive behaviours, are motivated to prevent infection, and perceive themselves as being capable of applying the recommended preventive strategies. In our first meetings with the community group we explored the needs and views of participants regarding a practical, action-based approach to HCV risk reduction. Based on the first feedback round we set the following IMB-inspired goals: on the information and motivation levels we wished to provide detailed HCV risk information and information on practical risk reduction steps to avoid HCV transmission and increase the response efficacy (i.e. increase a person's belief that the recommended steps will be effective in avoiding an HCV infection) and hence improve attitudes towards preventive behaviours. On the skills level, we wished to provide (illustrated) tips on how to perform such steps to increase perception of self-efficacy an provide the actual products that can be used to perform the suggested steps. We subsequently held several sessions about the actual toolbox content and supporting instructional materials. During these sessions it became clear that many community members were uncertain about how to effectively clean and disinfect their toys, hands and other surfaces, and they voiced their need for cleaning and disinfection instructions. This resulted in adding a number of cleaning and disinfection materials and instructions to the toolbox. An example of these instructions is the step-by-step explanation on how to disinfect hand and forearms: 1) wash with soap to remove all lube from your skin and rinse well; 2) dry your skin, 3) make a cup with one hand and fill it with hand disinfectant; 4) disinfect hands and forearms by rubbing hand disinfectant for at least 30 minutes. Another example is how to disinfect sex toys and anal douches which was explained in 4 steps: 1) Clean sex toys and anal douche with dish-washing liquid to remove all lube and rinse well; 2) mix 1 part bleach with 9 parts cold water; 3) submerge for 5 minutes in the bleach solution: 4) rinse thoroughly with water and wipe dry with a clean towel.

Decisions on the addition of other products to the toolbox were collectively made with the community (Fig 1). Consensus was easily reached for most of the products except for the safe drug use equipment. The suggestion to add needles and syringes to the toolbox led to an extensive discussion within the community group. Some members were in favour of including safe drug use equipment as drug use is a reality in their environment. Whilst other members

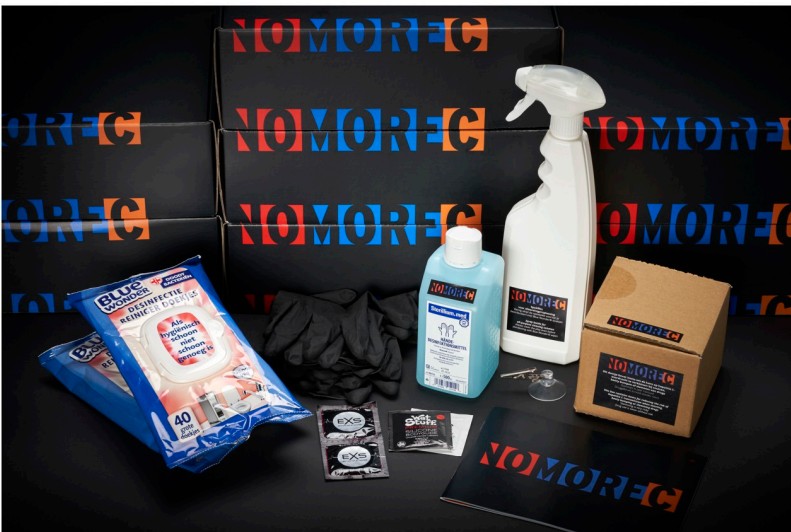

**Fig 1. NoMoreC toolbox.** The box contains a spray bottle (to clean surfaces where sexual activity takes place), sterilium® med hand disinfectant, hydrogen peroxide cleaning wipes (to disinfect surfaces where sexual activity takes place), gloves (for safe fisting), condoms (for safe sex), disinfection instruction card for hands and forearms, disinfection instruction card for dildos, toys & anal douche, a safe drug use box and booklet with information about HCV risk reduction. Republished from www.NoMoreC.nl under a CC BY license, with permission from the NoMoreC project group, original copyright 2018.

believed that injection equipment would be experienced as shocking or may trigger drug use. However, because safe drug use is essential to prevent HCV among men who engage in chem-sex, a compromise was reached and the safe drug-use equipment was included in the toolbox but packaged in a separate sealed off box, so that the injection equipment was not directly visible when opening the toolbox. After the contents of the toolbox was decided upon, items were ordered, and all boxes were assembled and packed by the community members. A total of 700 toolboxes were packed; 450 in 2018 and an additional 250 in 2019. The final version of the toolbox contained an information booklet and referrals to the NoMoreC website where supporting information, testimonials and instructional videos were available. The toolbox also contained condoms, fisting gloves, disinfectants, equipment for safe drug use and a booklet with practical information on HCV risk reduction (Figs 1 & 2). Table 1 shows how the IMB model constructs have been integrated in the intervention.

From March 2018, the toolbox could be ordered online free of charge from the NoMoreC project website (www.nomorec.nl/en/get-yourself-a-toolbox/). They were sent to addresses in The Netherlands only. Toolboxes were also available for pick-up at 17 locations in Amsterdam: at four fetish shops, a condom shop, the city central STI clinic of the Public Health Service, six HIV outpatient treatment centres, four GP practices, and a non-governmental organisation (NGO) dedicated to improving the health and quality of life of people who use drugs (PWUD). Health care professionals were given the toolboxes to both distribute to MSM at risk and to discuss HCV prevention strategies with their clients.

## Study design, participants and procedures

A mixed-methods study was conducted among toolbox recipients, using quantitative and qualitative data collection methods. Study participants of both the quantitative and the qualitative studies were included if they were at least 18 years of age, were a man who had sex with

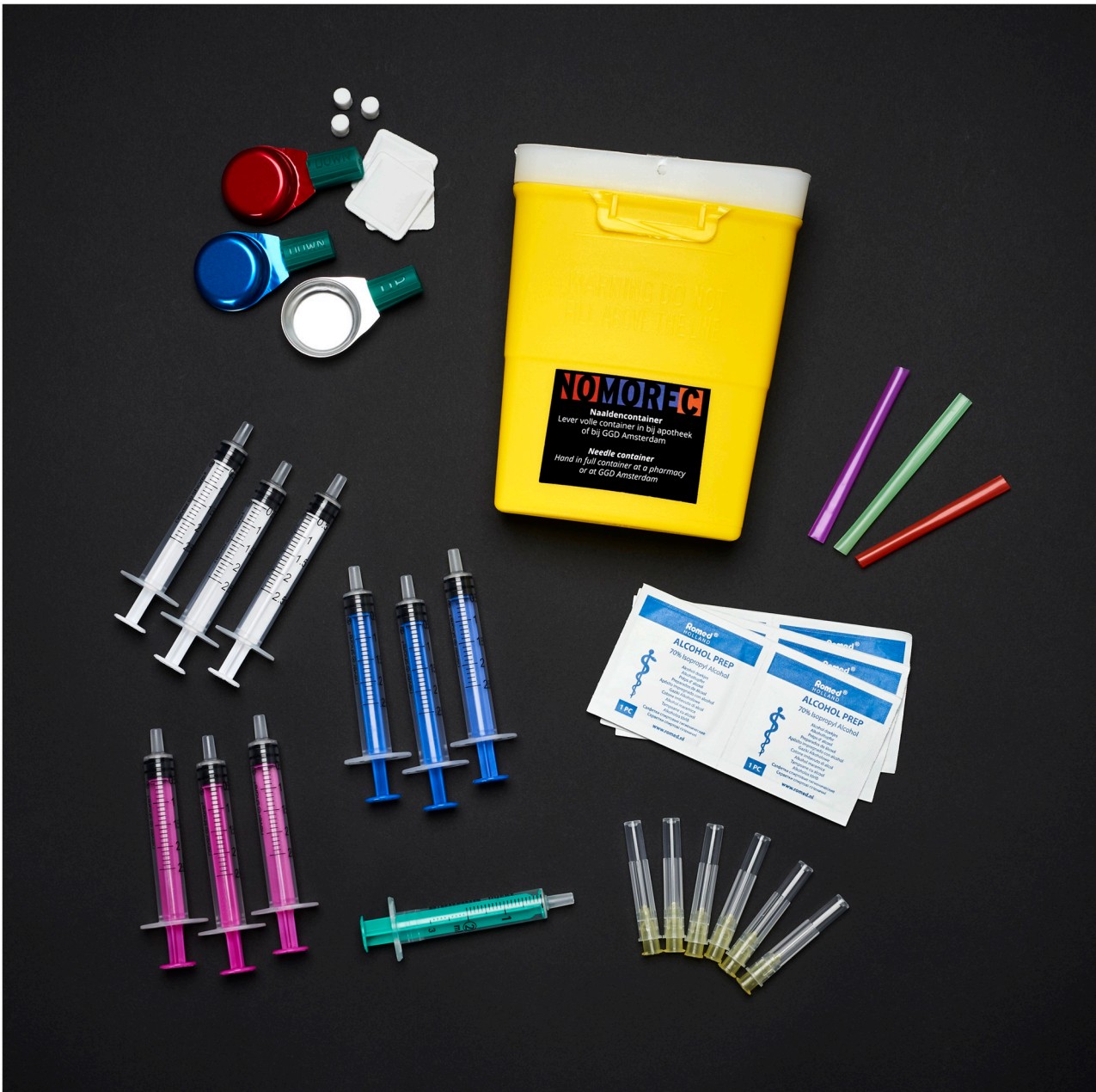

**Fig 2. Safe drug use box.** A small cardboard box containing: snorting straws, a sharps container, syringes, needles, stericup® mixing cups and alcohol wipes. A sticker with the text "Never share drug-use equipment. Pick your own colour" is stuck on the box. Republished from www.NoMoreC.nl under a CC BY license, with permission from the NoMoreC project group, original copyright 2018.

men, had received the toolbox and spoke the Dutch or English language. Firstly, online questionnaires were sent to toolbox recipients by email, 12 weeks after receipt of the toolbox in the period between March 2018 and March 2020. The questionnaire contained questions about HCV infection concerns, reasons for obtaining the toolbox, its use, context of use and impact of the toolbox on HCV risk reduction awareness (S1 Appendix). Answer options included yes/no, (3 and 5 point) Likert scales and lists with pre-defined answers (see S1 Appendix for

**Table 1. Toolbox intervention components focussing on information, motivation and behavioural skills constructs and their aims.**

| Toolbox component | IMB-construct | Aim |
|---|---|---|
| **Information booklet** | Information | Increase knowledge about: hepatitis C and its transmission, personal HCV risk, practical HCV prevention steps and their efficacy (response efficacy). |
| | Motivation | |
| | Behavioural skills | |
| Referral to **instruction videos** on the NoMoreC website | Behavioural skills | Teach HCV prevention skills; |
| | | Enhance self-efficacy to engage in HCV risk reduction behaviours |
| Referral to personal video **testimonials** about what men at risk do to reduce their HCV risk | Motivation | Develop positive attitude toward HCV risk reduction, improve motivation to engage in HCV prevention behaviours |
| **Sterilium® med disinfectant** for disinfection of hands and forearms | Motivation | Lower threshold to use the product |
| **Disinfectant cleaning wipes** for disinfection of surfaces where sexual activity takes place | Motivation | Lower threshold to use the product |
| **Spray bottle** for cleaning surfaces where sexual activity takes place before disinfection | Motivation | Lower threshold to use the product |
| **Black latex gloves** for safe fisting | Motivation | Lower threshold to use the product |
| **Black nitrile gloves** for safe fisting | Motivation | Lower threshold to use the product |
| **Black condoms** for safe anal sex | Motivation | Lower threshold to use the product |
| **Illustrate instruction card**: How to disinfect dildos, toys and anal douche | Behavioural skills | Teach disinfection skills; |
| | | Enhance self-efficacy to engage in this HCV risk reduction step |
| **Illustrated instruction card**: How to disinfect ands and forearms | Behavioural skills | Teach disinfection skills; |
| | | Enhance self-efficacy to engage in this HCV risk reduction step |
| **Clip** with suction cup to hang the instruction card for disinfecting hands and forearms on bathroom mirror | Motivation Behavioural skills | Lower threshold to use the product |
| **Safe drug use box** (Fig 2) for safe chemsex | Motivation | Lower threshold to use the product |

complete questionnaire and answer options). Two members of the community group pre-tested the questionnaire and were asked to give feedback about the clarity of the questions and answer options. After the pre-test, some small revisions to the questionnaire were made. The validity of the questionnaire was not pre-tested. The questionnaire provided data that were subsequently used as the basis for the qualitative study, to further explore the themes of the questionnaires, build our understanding of the experiences of toolbox recipients and give more meaning to the quantitative findings.

The qualitative part of the study consisted of in-depth interviews with men who had received the toolbox from March 2018 onwards. Men who had ordered the toolbox online were given the option to leave their email address if they were willing to share their opinion about the toolbox in the future. Men who had received the toolbox in person from a health care professional or had picked it up at a fetish shop were asked if they could be contacted in the future, to share their experience with the toolbox. Those who agreed to be contacted left their email. In February 2020, all toolbox recipients who agreed to be contacted were sent an email outlining the research aims and request for participation in the study. 20 men reacted to the email and agreed to participate in the study. In the period March till May 2020, 15 out of 20 participants who had given their consent took part in an in-depth telephone interview, after which thematic saturation was reached. The interviews lasted 30 to 60 minutes. A topic guide (S2 Appendix) with open ended questions was used, to give participants the opportunity to describe their personal experiences with the toolbox. During the interviews, topics discussed included: experiences with hepatitis C, reasons for obtaining the toolbox, the actual use of the different toolbox products, motivations for, and barriers to the use of the toolbox products, context of use and experiences with the use of the products. We asked participants to reflect on the possible impact of the toolbox on their HCV awareness and on the participants' sexual

and drug-use behaviours. We were interested to learn if participants had made any behavioural changes in their sex lives to reduce the HCV infection risk, based on the information in the toolbox, and what these changes were. For example, did participants start cleaning and disinfect dildos and the area where sex takes place with the suggested products? Did participants who use drugs, safely administer their drugs by single use drug equipment and not share their equipment? Did men who enjoy fisting start using gloves and clean and disinfect their hands and forearms immediately after fisting? Furthermore, we asked the participants if the information provided by the toolbox assisted in making a personal HCV risk assessment. To thank the participants for their contributions they received a gift voucher of €20.

In addition, professionals from the STI clinic, HIV centres in Amsterdam and NGO for PWUD were contacted by email in February 2020, requesting their participation in our study. In March and April 2020, 7 out of 8 professionals who consented to be contacted took part in an in-depth telephone interview, after which thematic saturation was reached. The interviews lasted 10 to 30 minutes. Again a topic guide with open ended questions was used, to give professionals the opportunity to describe their experiences with the toolbox in their work setting. During the interviews, topics discussed included: their professional experience with hepatitis C, use of the toolbox in their daily practice and triggers and barriers to using the toolbox. Interviews were audio-recorded, transcribed verbatim and anonymised.

## Data analysis

Simple descriptive statistics were used to analyse the quantitative data.

Interpretation of the qualitative data followed an inductive thematic analysis approach [19]. Two researchers (TP/JI) coded the transcripts independently and organized the data into open codes reflecting the major categories of information related to the study. Subsequently, codes were categorised into themes. Following the independent assessments of the transcripts, the researchers met frequently to compare and refine existing codes until they reached consensus on main themes and codes. MaxQDA Plus 2020 software (Release 20.0.8) was used for the qualitative data analysis.

## Ethics

Participants gave their informed consent by clicking the consent box prior to completing the online questionnaire. Participants, who were interviewed gave their informed consent verbally, which was recorded on audiotape and documented by transcription. We received exemption from the Amsterdam University Medical Centre's medical ethical committee for extended protocol review (Reference number: W20_110 # 20.145).

## Results

### Participants

**Toolbox recipients.** A total of 174 men were sent an email with a link to an online questionnaire between March 2018 and March 2020. The questionnaire was started by 65 participants and completed by 49, resulting in a response rate of 37% (65/174) and a completion rate of 75% (49/65). Median age of participants was 47 years (IQR 40–57), 100% were MSM living in the Netherlands (Table 2). The majority was born in the Netherlands (88%). More than a third reported they were HIV negative (n = 18, 37%), of whom 78% were using PrEP. Twelve participants (24%) reported to have had an HCV infection in the past, of whom one participant reported more than one infection. The majority (83%) of men who reported a past HCV infection were HIV positive. Fourteen participants (29%) were worried about getting HCV infected, 22 (45%) were somewhat worried and 13 (27%) were not worried.

**Table 2. Characteristics and concerns of toolbox recipients who filled out the online questionnaire (March 2018 and March 2020) or took part in an in-depth telephone interview (March-May 2020).**

|  | Online questionnaires (n = 49) | Interviews (n = 15) |
|---|---|---|
| **Socio demographic characteristics** | | |
| Age, years | 47 (40–57) | 48 (41–59) |
| Born in the Netherlands | 43 (88%) | 14 (93%) |
| Educated to college degree or higher[a] | NA | 13 (87%) |
| Employed[a] | NA | 12 (80%) |
| **HIV and HCV status (self-reported)** | | |
| HIV status: positive/negative/ not disclosed | 29 (59%) / 18(37%) / 2 (4%) | 8 (53%) / 7 (47%) / 0 (0%) |
| Current PrEP use | 14 (78% of HIV-negative MSM) | 6 (86% of HIV-negative MSM) |
| Past HCV infection: Primary/ Re-infection | 11 (22%)/ 1 (2%) | 5 (33%)/ 0 |
| Combined: | | |
| Past HCV (re)infection and HIV positive | 10 (20%) | 4 (27%) |
| Past HCV infection and HIV negative | 2 (4%) | 1 (7%) |
| **HCV-related concerns** | | |
| I worry about getting hepatitis C | 14 (29%) | 4 (27%) |
| I worry somewhat about getting hepatitis C | 22 (45%) | 7 (47%) |
| I do not worry about getting hepatitis C | 13 (27%) | 4 (27%) |

Data are in n (%) or median (IQR). PrEP = Pre-exposure prophylaxis. NA = not applicable.

[a]:Only asked during the in-depth interviews.

All 15 participants who took part in an in-depth telephone interview were MSM; 14 were living in the Netherlands and one abroad. All but one were born in the Netherlands. The median age was 48 years (IQR 41–59; Table 2). Most participants had at least a college education (87%) and were employed (80%). Almost half (47%) were HIV negative. Of the HIV-negative MSM, 86% were using PrEP. Five participants (33%), of whom 4 were HIV-positive, reported an HCV infection in the past. Four participants (27%) were worried about getting HCV infected, 7 (47%) were somewhat worried and 4 (29%) were not worried.

**Health care professionals.** Seven professionals took part in an in-depth interview. Two professionals worked at the STI clinic and four professionals at an HIV treatment centre in Amsterdam: five as registered nurses and one as nurse practitioner. One professional worked for an NGO dedicated to improving the health and quality of life of PWUD. The median age was 56 (IQR 44–58). All participants regularly saw MSM at risk of HCV in their care settings. They offered the toolbox to their clients and referred them to the NoMoreC project website. Professionals reported to have used the toolbox to discuss HCV risk and reduction during the consultations.

We present the combined results of the quantitative and qualitative analyses under four main themes: 1) Reasons for obtaining toolbox; 2) Toolbox usability and acceptability; 3) Context of use of the toolbox; and 4) Self-reported impact of the toolbox on HCV awareness and behaviour change. For each theme we commence with the questionnaire results and subsequently offer more in-depth related perspectives from the qualitative data. Perspectives of health care professionals are integrated into theme 1 (Reasons for obtaining toolbox).

## Theme 1: Reasons for obtaining toolbox

**Participants.** The following reasons for getting a toolbox were given by 49 respondents of the online questionnaire: 35 (71%) wanted to reduce the risk of getting infected with HCV, 20

(41%) wanted to reduce the risk of transmitting HCV to someone else, 28 (57%) were curious to know the contents of the toolbox, 2 (4%) wanted to give the toolbox to someone else, 27 (55%) wanted to use it during sex parties and 6 (12%) got the toolbox because it was recommended to them.

During the in-depth interviews interesting paths and reasons for obtaining the toolbox were revealed. Exposure to information about HCV infection risk and the toolbox was sufficient to persuade men to try and use the toolbox:

> "I was looking into PrEP. I wanted to start using that and my GP advised me to investigate what that involved. So then I came across the toolbox. That was the moment that I realised: yeah, I am at serious risk (of HCV). I didn't even know it existed. I only knew of hepatitis A and B and I got vaccinated for that."

> "*So when you were informing yourself about PrEP and getting PrEP care, you learned about the NoMoreC toolbox. Is that correct?*"

> (interviewer)

> "Yes, that's correct."

> (participant 3)

Some participants mentioned they wanted to inform their friends and sexual partners and openly talk about their sex lives in relation to HCV risk. They felt the toolbox could assist them in discussing these topics.

> "You can reduce risk by openly discussing your sex life with friends. Or when you have a sex date or go to a place where you can have sex and just talk to like-minded people about it (HCV risk reduction). That box helps me a lot with that. . ... For example, I have shown the toolbox to a friend I regularly have sex with. I told him what the doctor had told me about all the products in the box. We then had a nice discussion about it."

> (participant 6)

**Health care professionals.**   Health care professionals indicated that the toolbox supports them in discussing HCV risk and HCV risk reduction. They want to inform their clients about HCV infection and how to prevent it. One professional explained:

> "I always ask first: "We pay attention to hepatitis C, shall I tell you more about that? Would you like me to show you something?" And then I show the box. I tell them that they can have the box. But I always first show what it contains and say: 'Some things apply to you, other things do not apply to you, but this is all aimed at preventing hepatitis C or reducing the chance of getting hepatitis C."

> (professional 2)

Some professionals mentioned that they showed the NoMoreC website first, including the risk reduction video's and subsequently explained the use of the toolbox products. Many professionals used a client centred approach when discussing HCV risk, based on the client's knowledge about HCV and sexual behaviour. A lack of HCV knowledge, high risk sexual behaviour and a past HCV-infection were motives for introducing the toolbox:

"During a consultation I find out what men know and what they don't know and how sexually active they are. Some say, 'No, a box like that is not for me' and others ask for more information. When I notice that someone is really interested, I tell them they can have the toolbox. For many men it (the toolbox) is an eye-opener and it also helps me discussing behaviour and risks. It helps because you can give very practical information."

(professional 4)

Some professionals indicated certain barriers for using the toolbox during consultations. The medical history form that is used during a consultation at the STI clinic is not inviting for introducing the toolbox and discussing settings in which it can be used as questions about group sex or visiting sex parties are not included. Several professionals also mentioned that sexual techniques such as fisting or sharing toys can be difficult to bring up during a consultation. Building a relationship with a client and seeing them more regularly, lowered this barrier and in some cases clients would bring up their sexual preferences more easily, as one nurse reported:

"Sometimes someone says: 'I like fisting, I go to fisting parties'. Then I will ask: 'have you heard of hepatitis C?' Also, I have information about hepatitis C on my table, so some men will ask about that and then it's an easy entry point to talk about it. But I don't just ask: 'Do you practice fisting?' or 'Do you share toys?'"

(professional 2)

Another professional mentioned that he did not have enough time to extensively discuss HCV risk reduction and explaining all the items in the toolbox. An HIV nurse explained that her focus is on HIV treatment and less so on HCV prevention:

"Well, I think it is a very handy box. It is just that our consultations are mainly focused on HIV treatment and if we notice that there is risk behaviour, we will of course discuss it. It just doesn't happen very often that we go into that matter (hepatitis C risk reduction) in more depth. So, it is not in my system to introduce that box."

(professional 7)

The use of the toolbox in the context of sexualised drug use was a problem for the professional who worked for an NGO dedicated to improving the health and quality of life of PWUD because he works with men who have recently stopped or want to stop engaging in chemsex. He was concerned that the safe drug use equipment in the toolbox would serve as a trigger for his clients who were battling addiction. Therefore, he did not hand out the toolbox to this group.

## Theme 2: Toolbox usability and acceptability

Of 49 online respondents, 38 (78%) respondents used one or more toolbox products in the past 12 weeks and 41 (84%) intended to use products in the future. The use of the individual toolbox products since receipt of the toolbox is presented in Table 3.

The great majority of the participants (46/49, (94%)) rated the instructions of the toolbox as clear or very clear, two participants rated them as neither clear nor unclear and one participant reported to have not read the instructions. Responses to other questions related to toolbox usability and acceptability are shown in Table 4.

During the in-depth interviews, the use of the products were typically mentioned in relation to the sexual technique and practices of the participant. All toolbox recipients had used or

**Table 3. Use and intended future use of the individual products of the HCV prevention toolbox among 49 online respondents.**

| Toolbox product | Product use Number of participants who used the product, n (%) | Future product use Number of participants who intend to use the product in the future, n (%) |
|---|---|---|
| **Disinfectant wipes** | 28 (57%) | 41 (84%) |
| **Sterillium® med hand disinfectant** | 25 (51%) | 32 (65%) |
| **Hand & lower arm disinfection instruction card** | 23 (47%) | 17 (35%) |
| **Condoms** | 17 (35%) | 16 (33%) |
| **Syringes for rectal administration of drugs** | 15 (31%) | 19 (39%) |
| **Sharps container** | 15 (31%) | 17 (35%) |
| **Latex gloves** | 14 (29%) | 25 (51%) |
| **Spray bottle** | 12 (24%) | 27 (55%) |
| **Nitril gloves** | 11 (22%) | 15 (31%) |
| **Syringes and needles for injecting drugs** | 11 (22%) | 14 (29%) |
| **Stericup® mixing cups** | 6 (12%) | 11 (22%) |

Number of participants who used (product use) and intend to use (future use) the product is given per product, and percentage of the total respondents (n = 49).

tried the toolbox products; some used many and others used few products, depending on their personal HCV risk. Participants continued to use the products and some had ordered several products again. It was highlighted that some products were easier to use than others. Products that were received positively, were the disinfectant wipes, hand disinfectant and gloves. They were widely used and were highly accepted.

> "The disinfectant wipes, I'm very happy with those. They are used everywhere. I now have them in my suitcase, always"
>
> (participant 6)
>
> "That (sterilium® med) disinfectant is really nice. It doesn't stink. I use it especially when I am done fisting. Then I wash my arms and use it."
>
> (participant 1)
>
> "Those black gloves have been a really good tip, I still use them when necessary."
>
> (participant 3)

Mixed reactions were given about condoms and the safe drug use box. While some were happy the toolbox contained these products, others did not use them or disapproved of their inclusion.

> "I used the condoms, I always have a box of condoms."
>
> (participant 3)
>
> "To be honest, I haven't used the condoms very often."
>
> (participant 13)
>
> "I have given away the condoms, because I don't use those. I use a different brand of condoms"
>
> (participant 8)

**Table 4. Measures of toolbox usability and acceptability.**

|  | Number of participants (%) |
|---|---|
| **Usability** | |
| The instructions for the items in the toolbox were: | |
| Very clear | 21 (43%) |
| Clear | 25 (51%) |
| Neither clear nor unclear | 2 (4%) |
| I haven't read the instructions | 1 (2%) |
| The instructions for disinfection were: | |
| Very clear | 21 (43%) |
| Clear | 24 (49%) |
| Neither clear nor unclear | 1 (2%) |
| Unclear | 1 (2%) |
| I haven't read the instructions | 2 (4%) |
| Used the toolbox to start a conversation about hepatitis C with: | |
| Friends and sex partner(s) | 18 (37%) |
| Friends (s) only | 4 (8%) |
| Sex partner(s) only | 5 (10%) |
| I have not used the toolbox to start a conversation | 11 (22%) |
| No response | 11 (22%) |
| **Acceptability** | |
| It is good that a box with a sharps container and drug equipment is included in the toolbox | |
| Strongly agree | 25 (51%) |
| Agree | 14 (29%) |
| Neither agree nor disagree | 9 (18%) |
| Strongly disagree | 1 (2%) |
| I would recommend the toolbox to a friend or sex partners | |
| Definitely | 34 (69%) |
| Probably | 11 (22%) |
| Maybe | 3 (6%) |
| Probably not | 1 (2%) |

Responses of 49 participants who completed the online questionnaire 12 weeks after having received the toolbox.

"It (the safe drug use box) is of added value. The only thing I haven't used are the mixing cups. All the other things I have used."

(participant 7)

"I don't use (drugs) myself, but I know friends use. So it is nice to have."

(participant 1)

".. it contained needles and that sort of stuff. I was bit shocked by that. On the one hand, I think it is really good that it is included, but on the other hand I think: are you not promoting something?"

(participant 13)

### Theme 3: Context of use of the toolbox

Of 49 online respondents, 28 (57%) used the toolbox during one-on-one sex, 22 (45%) used it during group sex with 3 to 4 men or at small sex-parties. Few respondents, 3 out of 49 (6%), reported the use of the toolbox at bigger sex parties with 5 men or more. One respondent used the toolbox in a group sex setting with both men and women and one respondent used it during tantric massages. A third of the participants (16/49 (33%)) discussed the use of the toolbox products with their sexual partner(s) before having sex, 10/49 (20%) discussed it with some of their sex partners, 12/49 (25%) did not discuss it and 11/49 (22%) participants did not answer the question.

During the in-depth interviews, all participants mentioned that they used the toolbox with sexual partner in a home-setting. The following participants described in what context they informed their sexual partners of the toolbox:

"I just use the toolbox during sex. This can be one-on-one or in a group. I always have the disinfectant to clean your hands after having played. And I tell people: 'Here is the dishwashing liquid and here is the disinfectant. This card shows how to clean your hands'."

(participant 5)

"When I organize a party, I just put that box in the room and say: 'Gentlemen, all kinds of tools for a safe party'."

(participant 6)

One participant described the use of the toolbox in the context of sexualized drug use:

"We slam (inject drugs) a lot. The fun thing is, the toolbox contains everything that you need to prevent needles and stuff being exchanged, because of those different colour syringes from the toolbox, everyone uses his own colour. So I put everything on the kitchen table with a note so everyone knows: this is my colour. Because of the toolbox we have started to work very systematically"

(participant 7)

Some participants expressed doubts in their ability to consistently apply the risk reduction strategies promoted by the toolbox, to their sex lives. It was felt that the products were easy to use in a home-setting but when having sex outside their home it was more difficult, as nobody would take the toolbox to a club or sauna. However, some reported that they compensate that by taking individual products when going out or on holidays.

"I don't take the toolbox to a club, that's a no brainer. But I pack some things in my suitcase when I go to a festival or when I travel, like gloves and hand disinfectant. But never the whole box, that is too much luggage."

(participant 6)

It was also reported that the use of the products was easier if the sex partner had seen the toolbox before and was already aware of the products.

## Theme 4: Self-reported impact of the toolbox on HCV awareness and behaviour change

Of 49 online respondents, 23 (47%) strongly agreed, 23 (47%) agreed and 3 (6%) did neither agree nor disagree with the statement: "The NoMoreC toolbox gives me a better understanding of what I can do to reduce the risk of hepatitis C".

During the interviews, participants described that the toolbox had contributed to increasing their knowledge and awareness about hepatitis C and its transmission. They mentioned being able to better predict their risk of contracting hepatitis C and that the toolbox products helped them to reduce this risk. Both the toolbox products and being able to make different choices regarding the risks they took gave many participants a sense of safety as well as a feeling of being in control of their own health. The ability to protect others was also considered important. Participants reported that they had considered the preventive measures suggested in the toolbox, and implemented those that were attainable for them. Small changes in their behaviour were mentioned, such as the consistent use of disinfectant wipes or cleaning of sex toys.

> "I used to clean things in a certain way, but dildos for example, you are supposed to put them in a bucket of water with bleach. I used to clean them with soap and then disinfect them. But with bleach it's better, because you get into the pores. . ..so I know that now, and put the dildos in a bleach solution before I use them, so they are clean."

One participant mentioned that since he had implemented the new hygienic measures and use of disinfectants that friends felt safe to join his sex parties. Others said that since exposure to the toolbox and being better informed about HCV risks, they had made a conscious decision to stop visiting sex parties. They felt by having one-on-one sex only, they could better control their risk:

> "My sexual behaviour has changed. I don't go to big parties anymore. I prefer one-on-one, because I think that protects me more than the measures during sex.
>
> *When you say: 'I have changed my behaviour', have you done that, based on the information in the NoMoreC toolbox?*
>
> (interviewer)
>
> Yes, it has contributed of course. I have also visited the website a few times to inform myself. And you know, at the dates I have had in the past 2 years [after toolbox reception], I have always asked: 'How do you deal with hepatitis C?'."
>
> (participant 9)

Next to the practical use of the toolbox products, the toolbox was also felt to provide good information to assess HCV risk and aid discussions about hepatitis C and risk reduction:

> "That box helps a little to get a conversation going about hepatitis C, collect information and make a risk assessment. Especially in combination with the website
>
> *Can you explain how the toolbox helps you to get a conversation going?*
>
> (interviewer)
>
> "Well, someone will see the toolbox in my house and say: 'Hey, did you also get that box?' and then we will start talking about it. It sticks in people's minds when they have had that

box, seen it or spoken about it. I hear from some of my friends: 'I have had that box and I certainly learned something from it', we talk about the things (toolbox products) we use."

(participant 2)

Participants recommended or had given the toolbox to a friend and had used it to educate friends and sexual partners on risk reduction strategies. It was mentioned that there was still a lot of ignorance and lack of knowledge about hepatitis C among MSM.

"The good thing about that box is that I have talked about it with several friends, who didn't know anything about hepatitis C. And when I talked about hepatitis, they thought I was talking about A and B and said: 'I am vaccinated for that'. So, I thought it was a good thing that the toolbox got that conversation started."

(participant 13)

## Discussion

Our study adds to the understanding of how MSM at risk of HCV and health care professionals respond to an innovative HCV risk reduction intervention such as the NoMoreC toolbox. The mixed-methods approach used, which combines quantitative and qualitative data, offers a valid and deeper understanding of participants' experiences with the toolbox and its possible impact on HCV awareness and behaviour change. We show that the toolbox gave men a sense of safety and control by being more aware of their HCV risks and by providing easier access to the right protection tools for themselves and others. Data collected from professionals, indicated that the toolbox is a useful aid to discuss HCV risk and risk reduction with clients.

The main motivation for MSM to obtain the toolbox is to reduce the risk of getting an HCV infection. All participants used the toolbox products to some degree and indicated good usability of the toolbox and found the instructions clear. Some participants indicated that the provision of the products has lowered the threshold for them to use these products, and facilitated their risk reduction behaviour. Furthermore, men reported the intention to use some of the products in the toolbox in the future. The use of the toolbox to openly discuss one's sex life, hepatitis C risks and related risk reduction strategies with friends and sexual partners was reported by many participants and suggests that the toolbox has contributed to encourage men to discuss sensitive HCV related topics. It has previously been shown that peers are more likely to influence behaviour, compared to mass media programs, since they are able to build trust among fellow group members, which allows for open discussions on sensitive topics [20]. We believe that the 'viral' quality of the toolbox can contribute to further inter-peer dissemination of HCV knowledge and awareness. In the field of HIV prevention it has also been shown that peer interaction is an effective tool with long term effect for behaviour change among groups at high-risk of HIV [21].

The personal experiences with the toolbox demonstrate that the impact is different for each user. Some users reported that the NoMoreC toolbox has mainly impacted their knowledge and awareness about HCV, risk reduction and has helped them to assess their personal HCV risk. Others explained that they have changed their sexual behaviour based on the information provided. The use of the toolbox as a conversation starter was appreciated by many men, showing the wide acceptability of the toolbox within the target group. Furthermore, the users reported to use the toolbox products during one-on-one sex, group sex and chemsex, showing that the toolbox is suitable in a variety of contexts. Even though some dismissive reactions

from users were reported in relation to the safe drug use box in the toolbox, almost one third of the quantitative study participants uses items for safe drug use and almost 40% intended to use them in the future. This suggests that the provision of safe drug equipment is important to some of the men in our target group and was reported in some cases to facilitate more structural safe practices of drug use. We were not surprised to find that some users were somewhat shocked by the safe drug use box, because its inclusion in the toolbox caused the most discussion during the development process [14]. We believe that the provision of safe drug equipment is important to some men of the target group, and even though controversial it does answer to a need.

Our results show that among professionals of the STI clinic and HIV treatment centres in Amsterdam the toolbox is a well-received intervention. They reported that the toolbox is a good educational tool that assisted them in discussions about risk reduction with clients. However, for some professionals, bringing up certain sexual techniques such as fisting and sharing toys can be difficult. In addition, consultation protocols that do not offer time or space to thoroughly discuss high-risk sexual behaviour and risk knowledge hamper the possibility to identify men at risk of HCV. This was also highlighted by a Swiss study among HIV-positive MSM with a past HCV infection, which showed that if professionals rely only on condomless anal sex with non-steady partners as a criterion to identify men at risk of HCV infection, disregarding information on other high risk-behaviours, such as fisting or group sex, they miss a proportion of MSM at risk [22]. We recommend the use of a comprehensive list of potential risk behaviours to identify men whose sexual and drug use behaviours increase their risk of HCV infection. The toolbox can assist health professionals in identifying MSM at risk of HCV and prompt risk reduction conversations within consultation settings.

There are some limitations to this study. Participants of the quantitative study received the online questionnaire 12 weeks after the receipt of the toolbox, therefore we measured their experiences with the toolbox over a relatively short period. However, we did provide in our qualitative data the perspective of users who had received the toolbox up to two years prior to the interview, giving some insight into their experiences and possible behaviour change over a longer period of time. Future studies into the psychosocial, cultural, and personal factors that contribute to long term HCV risk reduction behaviour are recommended. In addition, support needs for MSM who are willing to change their behaviour and for MSM who are resistant to change are likely to be different. Therefore, studying both groups is essential for the development of future effective interventions.

We acknowledge that our study may have limited representativeness as data was collected only from participants who already ordered the toolbox and were willing to participate in the study, or from professionals who had reported use of the toolbox in the past. These participants may have been more positive about the intervention than those who did not want to obtain or use the toolbox, giving more favourable study results. Nevertheless, even among this highly motivated sample we uncovered some of the barriers to the (consistent) use of the toolbox.

## Conclusion

This study has illustrated how a practical and relatively simple intervention, might contribute to facilitating HCV risk reduction. The HCV prevention toolbox was well-received by both MSM at risk and professionals. It contributed to raising awareness of HCV risk, implementing risk reduction strategies and it has given men a better understanding of the risk factors and how to reduce these. Our study showed how the toolbox was used to disseminate inter-peer HCV-related information and suggest the potential of the toolbox approach in the effort to

impact the community norms. Furthermore, the toolbox can serve as an example intervention for countries that have similar sexual HCV transmission dynamics among MSM.

## Supporting information

**S1 Appendix. Questionnaire sent to toolbox recipients.**
(PDF)

**S2 Appendix. Interview guide.**
(PDF)

## Acknowledgments

We would like to thank all participants and professionals, who shared their experiences with the toolbox with us.

## Author Contributions

**Conceptualization:** Tamara Prinsenberg, Udi Davidovich.

**Data curation:** Tamara Prinsenberg, Joël Illidge.

**Formal analysis:** Tamara Prinsenberg, Joël Illidge.

**Methodology:** Tamara Prinsenberg, Udi Davidovich.

**Supervision:** Udi Davidovich.

**Writing – original draft:** Tamara Prinsenberg.

**Writing – review & editing:** Tamara Prinsenberg, Paul Zantkuijl, Maarten Bedert, Maria Prins, Marc van der Valk, Udi Davidovich.

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
