## [Decision Letter · Decision Letter 0]

29 Jul 2021

PONE-D-21-13933

A mixed methods evaluation of an innovative hepatitis C risk reduction intervention for men who have sex with men

PLOS ONE

Dear Dr. Prinsenberg,

Thank you for submitting your manuscript to PLOS ONE. After careful consideration, we feel that it has merit but does not fully meet PLOS ONE’s publication criteria as it currently stands. Therefore, we invite you to submit a revised version of the manuscript that addresses the points raised during the review process.

Please ensure that a thorough proof-read is conducted on the manuscript. Both reviewers have noted that the IMB model is mentioned but not discussed in depth. I agree with this and encourage the authors to elaborate on the inclusion of the model and integrate it better throughout the manuscript. 

We look forward to receiving your revised manuscript.

Kind regards,

Benjamin R. Bavinton

Academic Editor

PLOS ONE

Journal Requirements:

2. Please include additional information regarding the survey or questionnaire used in the study and ensure that you have provided sufficient details that others could replicate the analyses. For instance, if you developed a questionnaire as part of this study and it is not under a copyright more restrictive than CC-BY, please include a copy, in both the original language and English, as Supporting Information. If the original language is written in non-Latin characters, for example Amharic, Chinese, or Korean, please use a file format that ensures these characters are visible.

3. Please state whether you validated the questionnaire prior to testing on study participants. Please provide details regarding the validation group within the methods section.

4. Please include a copy of the interview guide used in the study, in both the original language and English, as Supporting Information, or include a citation if it has been published previously.

“This project was performed within the MC Free consortium. MC Free is funded by grants from Gilead Sciences, AbbVie, Janssen-Cilag, Merck Sharpe & Dohme, and Roche Diagnostics. The funders had no involvement in the study design, writing of the manuscript, and decision to submit the article for publication. “

 ”TP and MP report speaker fees and grants from Gilead Sciences, Merck Sharpe & Dohme, and AbbVie paid to their institute. MvdV's institute received unrestricted research grants and consultancy fees from AbbVie, Gilead, Johnson & Johnson, Merck Sharpe & Dohme, and ViiV Health Care.”

Reviewers' comments:

Reviewer's Responses to Questions

**Comments to the Author**

1. Is the manuscript technically sound, and do the data support the conclusions?

Reviewer #1: Yes

Reviewer #2: Yes

2. Has the statistical analysis been performed appropriately and rigorously? 

Reviewer #1: Yes

Reviewer #2: Yes

3. Have the authors made all data underlying the findings in their manuscript fully available?

Reviewer #1: No

Reviewer #2: Yes

4. Is the manuscript presented in an intelligible fashion and written in standard English?

Reviewer #1: Yes

Reviewer #2: Yes

5. Review Comments to the Author

Reviewer #1: A mixed methods evaluation of an innovative hepatitis C risk reduction intervention for men who have sex with men

This paper details the mixed-methods evaluation of a toolbox provided to MSM at risk of HCV. The methods, analysis and write-up are straightforward and the output is suitable to a range of audiences within academia, clinical settings and the voluntary sector which is great to see. I have a few comments to further strengthen this publication.

The authors state that the contents of the tool-box were collectively decided with community members. It would be really helpful to have a greater understanding of this process and the reasons which underpinned intervention design. Co-creation is an important approach and it would help the reader to understand how this worked and what the implications were.

A related point, the authors state the IMB model was used as an explanatory framework in describing how the toolbox was formed. It would be helpful to have a greater elaboration on this. Currently it is only mentioned briefly in the introduction and the discussion, and I think there needs to be greater reflection. For example, was there a theory of change or a logic model produced which used on this model to conceptualise how the toolbox could facilitate behaviour change? If so this should be described and presented, perhaps in supplementary information. I’m also not sure how well the results demonstrate the utility of IMB model in this application, more reflection on this in the discussion is required.

Finally, and this is more of a suggestion, the results from the healthcare providers seem to mirror those of the MSM who used the tool-box quite closely. I think these could be merged easily to create a more coherent narrative around the tool-box and its utility. This would also improve legibility.

I did not proof-read or look for minor grammatical errors.

Thank you for the opportunity to review this interesting work.

Reviewer #2: Thank you very much for the opportunity to review this manuscript. This manuscript aimed to assess the use of an innovative hepatitis C risk reduction intervention and the impact on behaviors among men who have sex with men. The topic is of great importance, however, the research and result interpretation part may require further adjustment. The comments are as follows:

1. Title: the authors referred this study to an evaluation of the intervention. The study seemed more like a presentation of the current implementation of the intervention rather than an outcome evaluation, as the latter may require more sophisticated research design to conduct a causal inference. It may be better to adjust the title to better fit the research objectives.

2. Introduction: the authors regarded toolbox as an innovative intervention. It may be more persuasive to add some background of this intervention that indicates how innovative it is by comparing it with prior interventions.

3. Methods: the authors reported that the toolbox had been designed according to the IMB model. “According to the model, HCV prevention information and motivation each have a direct effect on behavior as well as combined effect through behavioral skills (P4 Lines 83-85). Please further elaborate this sentence by specifying how the IMB model informed the intervention design, what was the specific direct effect of each tool and what were the combined effect of this toolbox.

4. Research design: the authors reported that the objective was to assess the use of an innovative hepatitis C risk reduction intervention and the impact on behaviors. I assumed that the authors mainly concerned the consumers in terms of intervention utilization rather than providers. If it is the case, I suggest the authors focusing on reporting findings regarding consumers attitude and behaviors. Some findings from the interviews with providers could be integrated into main themes, for instance, barriers for intervention utilization behaviors.

5. Methods: the authors conducted online survey and telephone interview, respectively. Were these two samples mutually exclusive to each other? If not, it may not be appropriate to conduct inter-group comparison.

6. Results: please move the descriptions on research methods (e.g. “the interviews lasted 30 to 60 minutes” P9 Lines 178-179) to the Methods section.

6. PLOS authors have the option to publish the peer review history of their article (what does this mean?). If published, this will include your full peer review and any attached files.

Reviewer #1: No

Reviewer #2: No

---

## [Author Response · Author response to Decision Letter 0]

13 Oct 2021

Response to reviewers’ comments

Title: A mixed methods evaluation of an innovative hepatitis C risk reduction intervention for men who have sex with men 

This document includes the reviewers’ comments and our response to each comment. The changes we have made in the manuscript in order to address these comments are detailed in our responses

As suggested by reviewer #2 (comment 1) the title is changed to: Usability, acceptability, and self-reported impact of an innovative hepatitis C risk reduction intervention for men who have sex with men: a mix methods study

Reviewer #1: 

This paper details the mixed-methods evaluation of a toolbox provided to MSM at risk of HCV. The methods, analysis and write-up are straightforward and the output is suitable to a range of audiences within academia, clinical settings and the voluntary sector which is great to see. I have a few comments to further strengthen this publication.

1. The authors state that the contents of the tool-box were collectively decided with community members. It would be really helpful to have a greater understanding of this process and the reasons which underpinned intervention design. Co-creation is an important approach and it would help the reader to understand how this worked and what the implications were.

We have added more information on the co-creation process, under ‘Toolbox development and distribution’ in the methods section. In short, it involved: 1) meetings to assess information needs, 2)discuss and generate the intervention ideas, 3) brainstorm sessions on the content of the toolbox, 4) final decision making on the toolbox content and 5)packing of the toolbox. (clean revised version: lines 95-123)

2. A related point, the authors state the IMB model was used as an explanatory framework in describing how the toolbox was formed. It would be helpful to have a greater elaboration on this. Currently it is only mentioned briefly in the introduction and the discussion, and I think there needs to be greater reflection. For example, was there a theory of change or a logic model produced which used on this model to conceptualise how the toolbox could facilitate behaviour change? If so this should be described and presented, perhaps in supplementary information. I’m also not sure how well the results demonstrate the utility of IMB model in this application, more reflection on this in the discussion is required.

The IMB-model was solely used to guide the development of the intervention, and not for evaluation purposes. We have explained the reason for using the IMB model, as the basis of the intervention in the introduction IMB- based interventions have been shown to be effective in preventing risky sexual behaviours among groups exposed to these interventions (lines 82-84). 

In addition, we have elaborated on how the IMB model constructs were integrated in the intervention. We have given the aims for each construct and how the different intervention components contribute to reaching these aims. (clean version lines 104- 109) The intervention components and their focus on information, motivation and behavioural skills have been specified in Table 1. 

The reviewer is correct that we did not use the IMB-model for the assessment of the effect of the intervention. We concentrated on users’ experiences with the toolbox and report on these under four main themes: 1) Reasons for obtaining toolbox, 2) Toolbox usability and acceptability, 3) Context of use of the toolbox, and 4) Self-reported impact of the toolbox on HCV awareness and behaviour change. 

3. Finally, and this is more of a suggestion, the results from the healthcare providers seem to mirror those of the MSM who used the tool-box quite closely. I think these could be merged easily to create a more coherent narrative around the tool-box and its utility. This would also improve legibility.

We combined the perspectives of the toolbox recipients and health care professionals by integrating the results of the health care professionals into theme 1 (reasons for obtaining the toolbox).

I did not proof-read or look for minor grammatical errors. 

The manuscript was proof-read by an English native speaker.

Thank you for the opportunity to review this interesting work.

Reviewer #2: 

Thank you very much for the opportunity to review this manuscript. This manuscript aimed to assess the use of an innovative hepatitis C risk reduction intervention and the impact on behaviors among men who have sex with men. The topic is of great importance, however, the research and result interpretation part may require further adjustment. The comments are as follows:

1. Title: the authors referred this study to an evaluation of the intervention. The study seemed more like a presentation of the current implementation of the intervention rather than an outcome evaluation, as the latter may require more sophisticated research design to conduct a causal inference. It may be better to adjust the title to better fit the research objectives.

We agree that our study does not measure causal inference but rather the self-reported impact of the intervention by its users. Our study is still an evaluation, but not an impact evaluation that aims to establish causality between the intervention and outcomes. We concentrated mainly on the usability, acceptability and the self-reported impact of the intervention by its user. Therefore, we have changed the title to: Usability, acceptability, and self-reported impact of an innovative hepatitis C risk reduction intervention for men who have sex with men - a mix methods study

2. Introduction: the authors regarded toolbox as an innovative intervention. It may be more persuasive to add some background of this intervention that indicates how innovative it is by comparing it with prior interventions.

To the best of our knowledge, there is only one other published HCV risk reduction intervention for MSM that also aims to reduce high risk sexual behavior. This intervention was developed in 2016 as part of the Swiss HCVree trial. The intervention is a counselling intervention for HIV/HCV co-infected MSM, to improve self-regulation of risks associated with high risk sexual behaviours and sexualised drug use. The counseling intervention was provided in combination with DAA treatment. We have taken a different approach by targeting MSM based on their sexual practices and not their HIV/HCV-status.

Furthermore, our intervention is innovative because: 1) there is a lack of interventions that go beyond promoting condom use alone, 2) it contains practical tools for HCV risk reduction, 3) was co-created with the gay community. We have added this to our introduction.

3. Methods: the authors reported that the toolbox had been designed according to the IMB model. “According to the model, HCV prevention information and motivation each have a direct effect on behavior as well as combined effect through behavioral skills (P4 Lines 83-85). Please further elaborate this sentence by specifying how the IMB model informed the intervention design, what was the specific v and what were the combined effect of this toolbox.

We used the IMB model as a theoretical framework to guide the development of the intervention. We did not use the model to assess the effect of the intervention, we were interested in the users’ experiences with the toolbox. In addition, see the response to comment 2 of reviewer 1.

4. Research design: the authors reported that the objective was to assess the use of an innovative hepatitis C risk reduction intervention and the impact on behaviors. I assumed that the authors mainly concerned the consumers in terms of intervention utilization rather than providers. If it is the case, I suggest the authors focusing on reporting findings regarding consumers attitude and behaviors. Some findings from the interviews with providers could be integrated into main themes, for instance, barriers for intervention utilization

We integrated the perspectives of the professionals in theme 1: Reasons for obtaining the toolbox.

5. Methods: the authors conducted online survey and telephone interview, respectively. Were these two samples mutually exclusive to each other? If not, it may not be appropriate to conduct inter-group comparison. 

The reviewer is correct, there could possibly be overlap. We are not able to correct for that overlap since the questionnaires were collected anonymously. Therefore, we decided it would be best to remove the comparison.

6. Results: please move the descriptions on research methods (e.g. “the interviews lasted 30 to 60 minutes” P9 Lines 178-179) to the Methods section.

Study period, number of participants and duration of interviews have been moved from the results to the methods section.

We have amended our Acknowledgements section and Funding Statement to:

Funding statement

This study was performed within the MC Free consortium. MC Free is funded by grants from Gilead Sciences, AbbVie, Janssen-Cilag, Merck Sharpe & Dohme, and Roche Diagnostics. The funders had no involvement in the study design, writing of the manuscript, and decision to submit the article for publication. TP and MP report speaker fees and grants from Gilead Sciences, Merck Sharpe & Dohme, and AbbVie paid to their institute. MvdV's institute received unrestricted research grants and consultancy fees from AbbVie, Gilead, Johnson & Johnson, Merck Sharpe & Dohme, and ViiV Health Care.

Acknowledgements

We would like to thank all participants and professionals, who shared their experiences with the toolbox with us.

Data availability

We have uploaded our data to Figshare, they can be access on:

https://doi.org/10.6084/m9.figshare.16676872.v1

---

## [Decision Letter · Decision Letter 1]

26 Nov 2021

PONE-D-21-13933R1Usability, acceptability, and self-reported impact of an innovative hepatitis C risk reduction intervention for men who have sex with men: a mixed methods studyPLOS ONE

Dear Dr. Prinsenberg,

Thank you for submitting your manuscript to PLOS ONE. The reviewers both agreed that the previous comments have been adequately addressed. However, one of the reviewers has made some suggestions/asked some questions that I believe will strengthen the manuscript. These are mostly along the lines of being at your discretion. Therefore, I provisionally accept this mansucript, but invite you to submit a revised version of the manuscript that addresses the new points raised by the reviewer. 

We look forward to receiving your revised manuscript.

Kind regards,

Benjamin R. Bavinton

Academic Editor

PLOS ONE

Journal Requirements:

Reviewers' comments:

Reviewer's Responses to Questions

**Comments to the Author**

1. If the authors have adequately addressed your comments raised in a previous round of review and you feel that this manuscript is now acceptable for publication, you may indicate that here to bypass the “Comments to the Author” section, enter your conflict of interest statement in the “Confidential to Editor” section, and submit your "Accept" recommendation.

Reviewer #2: All comments have been addressed

Reviewer #3: All comments have been addressed

2. Is the manuscript technically sound, and do the data support the conclusions?

Reviewer #2: Yes

Reviewer #3: Yes

3. Has the statistical analysis been performed appropriately and rigorously? 

Reviewer #2: Yes

Reviewer #3: Yes

4. Have the authors made all data underlying the findings in their manuscript fully available?

Reviewer #2: Yes

Reviewer #3: Yes

5. Is the manuscript presented in an intelligible fashion and written in standard English?

Reviewer #2: Yes

Reviewer #3: Yes

6. Review Comments to the Author

Reviewer #2: I appreciate for the authors' response and revision. They have addressed all my comments and I do not have further comments on this manuscript.

Reviewer #3: The following mixed-methods study provides perceptions of a toolkit to reduce high-risk sexual behaviours and high-risk behaviours for HCV infection among MSM and health professionals. I have provided some additional comments that I hope will be of benefit.

Introduction: “The NoMoreC toolbox contains practical tools, to encourage HCV

risk reduction in different settings” – Please provide some examples for an audience that is not in the HCV field.

Methods: Development of the toolbox with a “group of MSM”. Who are the group of MSM? Are they are community group? If yes, perhaps state this upfront to increase clarity.

Methods: “practical risk reduction steps” – Perhaps provide some examples of these for an international audience that is not in the field of HCV.

Methods: “Decisions on the addition of other products to add to the toolbox…” What were the products? (refer to the Figure here as this detailed is provided later). Also, approximately many boxes were packed?

Methods: “Study participants from both studies…” Please reference the studies here.

Methods: “Possible impact of the toolbox on their behaviour…” Please provide some examples. Are we referring to just sexual behaviours and/or drug use? Or both? What aspects of behaviour at the authors particularly interested in?

Methods: Was there a recruitment strategy for the qualitative interviews with MSM or was it just anyone who received a toolbox could participate. Please explain.

Methods: How many STI clinics were contacted for a qualitative interview? It states that the researchers reached thematic saturation following a discussion with 7 participants? Is it possible that the intention was to reach more representatives from the clinics? Were their attempts to reach clinic based in both urban and rural areas? What was the sampling/recruitment strategy? Were the clinics sent the toolboxes or did they request the toolboxes?

Methods: “Discrepancies between initial codes and themes…” Please modify this sentence to increase clarity. Discrepancies between the codes AND themes? Or discrepancies when coding?

Methods: The quantitative data analysis is clear. Please consider adding a bit more detail regarding the qualitative data analysis (e.g., could explain what inductive thematic analysis is or provide a couple of references for the reader).

Results: “At risk of hepatitis C”. Please modify to HCV to be consistent throughout.

Results: Theme 2 and Theme 3 both mention that the toolbox was not utilised outside of the home due to its size. Please consider placing into one section so as to not be repetitive.

Discussion: A health professional did not want to discuss the toolbox due to the safe injecting equipment. One of the strategies implemented by the community group was to have the safe injecting equipment in another box within the box. Did the health professional know this? How might the box be marketed in such a way that the ‘baby does not get thrown out with the bath water’? It serves an important purpose but might not be marketed at all if health professionals are uncomfortable with the harm reduction approach to drug use. Perhaps the creation of two boxes? Or MSM could order a personalised toolbox?

Discussion: Do the study authors intend to distribute the toolbox at saunas and clubs? (as mentioned in the Results, participants felt uncomfortable carrying the box outside of the home).

Discussion: “…sexual behaviour and risk knowledge form structural barriers that impeded the identification of men at risk of HCV.” This sentence is not clear. Please consider modifying.

Limitations: “possible behaviour change over a longer period of time”. How long is it expected that such an intervention would have an impact? Are future studies needed in this area? How can MSM be supported to reduce their risk of HCV infection long-term (peer-based organisations? Trusting therapeutic relationships with their providers?). Also, as mentioned by the authors, presumably individuals in this study would be open to changing their at-risk behaviours. What about persons who were not interested in changing their sexual risk behaviours? What might be some strategies to engage this group?

7. PLOS authors have the option to publish the peer review history of their article (what does this mean?). If published, this will include your full peer review and any attached files.

Reviewer #2: No

Reviewer #3: No

---

## [Author Response · Author response to Decision Letter 1]

4 Jan 2022

We have submitted a revised version of the manuscript that addresses the new points raised by one of the reviewers.

Thank you for the opportunity to strengthen our paper. 

We have uploaded the response to reviewers document with a description of how the suggestions of the reviewer were incorporated in the manuscript.

---

## [Editor Report · Decision Letter 2]

25 Jan 2022

Usability, acceptability, and self-reported impact of an innovative hepatitis C risk reduction intervention for men who have sex with men: a mixed methods study

PONE-D-21-13933R2

Dear Dr. Prinsenberg,

We’re pleased to inform you that your manuscript has been judged scientifically suitable for publication and will be formally accepted for publication once it meets all outstanding technical requirements.

Kind regards,

Benjamin R. Bavinton

Academic Editor

PLOS ONE

---

## [Editor Report · Acceptance letter]

9 Feb 2022

PONE-D-21-13933R2 

Usability, acceptability, and self-reported impact of an innovative hepatitis C risk reduction intervention for men have sex with men: a mixed methods study 

Dear Dr. Prinsenberg:

I'm pleased to inform you that your manuscript has been deemed suitable for publication in PLOS ONE. Congratulations! Your manuscript is now with our production department. 

Kind regards, 

on behalf of

Dr. Benjamin R. Bavinton 

Academic Editor

PLOS ONE